# Systematic Study of Gold Nanoparticle Effects on the Performance and Stability of Perovskite Solar Cells

**DOI:** 10.3390/nano15191501

**Published:** 2025-10-01

**Authors:** Sofia Rubtsov, Akshay Puravankara, Edi L. Laufer, Alexander Sobolev, Alexey Kosenko, Vasily Shishkov, Mykola Shatalov, Victor Danchuk, Michael Zinigrad, Albina Musin, Lena Yadgarov

**Affiliations:** 1Department of Chemical Engineering, Biotechnology and Materials, Faculty of Engineering, Ariel University, Ariel 40700, Israel; 2Department of Physics, Faculty of Natural Sciences, Ariel University, Ariel 40700, Israel; 3Central European Institute of Technology—Nanotechnology, Brno University of Technology, 16200 Brno, Czech Republic

**Keywords:** perovskite solar cell, plasmonic enhancement, gold nanoparticles, finite-difference time-domain (FDTD) simulations, inkjet printing, interface engineering

## Abstract

We explore a plasmonic interface for perovskite solar cells (PSCs) by integrating inkjet-printed TiO_2_-AuNP microdot arrays (MDA) into the electron transport layer. This systematic study examines how the TiO_2_ blocking layer (BL) surface conditioning, AuNP layer positioning, and nanoparticle loading collectively influence device performance. Pre-annealing the BL increases its hydrophobicity, yielding smaller and denser AuNP microdots with an enhanced localized surface plasmon resonance (LSPR). Positioning the AuNP MDA at the BL/perovskite interface (above the BL) maximizes near-field plasmonic coupling to the absorber, resulting in higher photocurrent and power conversion devices; these trends are corroborated by finite-difference time-domain (FDTD) simulations. Moreover, these devices demonstrate better stability over time compared to those with AuNPs at the transparent electrode (under BL). Although higher AuNP concentrations improve dispersion stability, preserve MAPI crystallinity, and yield more uniform nanoparticle sizes, device measurements showed no performance gains. After annealing, the samples with the Au content of 23 wt% relative to TiO_2_ achieved optimal PSC efficiency by balancing plasmonic enhancement and charge transport without the increased resistance and recombination losses seen at higher loadings. Importantly, X-ray diffraction (XRD) confirms that introducing the TiO_2_-AuNP MDA at the interface does not disrupt the perovskite’s crystal structure, underscoring the structural compatibility of this plasmonic enhancement. Overall, our findings highlight a scalable strategy to boost PSC efficiency via engineered light-matter interactions at the nanoscale without compromising the perovskite’s structural integrity.

## 1. Introduction

Perovskite solar cells (PSCs) have gained significant attention as a promising class of solution-processed photovoltaic technologies owing to their tunable bandgap, strong light absorption, and low-temperature fabrication [1]. Despite their rapid development, there is still considerable room for improvement in device performance and long-term operational stability. One key limitation arises from the thin nature of the perovskite absorber layer, which restricts light-harvesting efficiency due to incomplete absorption of the solar spectrum. For example, a typical methylammonium lead iodide (MAPI) film with a thickness of ~350 nm does not fully capture incident light, leading to optical losses [2,3]. Addressing this challenge requires new strategies that can enhance optical absorption without compromising charge transport or device structure.

One promising approach involves incorporating metallic nanostructures to exploit localized surface plasmon resonance (LSPR). In particular, gold nanoparticles (AuNPs) can enhance optical absorption via near-field enhancement, light scattering, and energy transfer mechanisms [3,4]. These plasmonic effects increase the effective optical path length within the active layer and amplify the local electromagnetic field, thereby improving exciton generation and charge carrier dynamics. Additional benefits may include hot electron injection and plasmon-induced resonant energy transfer [5,6,7]. A range of plasmonic integration approaches, including the incorporation of nanoparticles into charge transport layers and the formation of composite oxide architectures, has shown promise in enhancing the efficiency of photovoltaic devices [8].

In our previous work, we introduced a novel interface engineering technique by inkjet-printing a TiO_2_ microdot array embedded with AuNPs (TiO_2_-Au_MDA) atop the electron transport and hole-blocking layer (BL). This structure created photoactive microregions that enhanced photon trapping and charge extraction. Devices incorporating this layer achieved more than 46% improvement in PCE compared to reference cells without AuNPs, highlighting the potential of such plasmonic microstructures for improving device performance [9].

However, to achieve maximum benefit from the implementation of plasmonic microstructures, several key questions remain unresolved. First, the treatment of the compact TiO_2_ BL before printing plays a critical role in determining surface roughness and hydrophilicity, which affect the contact diameter of the droplet when it wets the substrate, then the evaporation process of the volatile components of the ink, and ultimately the size distribution of the gold nanoparticles. Optimizing BL treatment—particularly annealing—can significantly affect plasmonic behavior and overall performance [10].

Second, the spatial arrangement of layers, specifically the placement of AuNPs within the device stack, may influence charge recombination dynamics. While positioning AuNPs close to the perovskite layer enhances LSPR coupling, it may also increase the risk of charge trapping or recombination at interfaces [11,12]. In this study, we compare two PSC architectures and time stability: the AuNPs placed at the FTO/BL interface and the AuNPs positioned at the BL/perovskite interface. Through experimental analysis and finite-difference time-domain (FDTD) simulations, we evaluate how nanoparticle positioning affects photocurrent generation and overall device performance.

Finally, we address the role of AuNP concentration. While increased loading can improve LSPR intensity, it also raises the risk of aggregation, increased recombination sites, and reduced optical transparency. Conversely, concentrations that are too low may fail to produce meaningful enhancement. We systematically examined the influence of three AuNP concentrations on nanoparticle size distribution, absorption spectra, and photovoltaic behavior to identify an optimal formulation. Structural analysis further confirmed that varying AuNP concentrations did not alter the crystallographic phase or integrity of the MAPI layer, ensuring that optical and electrical properties were not influenced by structural degradation.

This study provides a comprehensive evaluation of AuNP-based plasmonic engineering in PSCs, focusing on the interplay between BL treatment, nanoparticle positioning, and AuNP concentration. The findings offer practical insights into optimizing both light absorption through plasmonic enhancement and efficient charge transport across a wide range of gold nanoparticle loadings. These results highlight the potential of interface-engineered plasmonic strategies to improve device performance while maintaining structural and electronic integrity.

## 2. Materials and Methods

Detailed descriptions of materials, synthesis procedures, and experimental methods are provided in the Appendix A.

## 3. Results

The standard perovskite solar cell (PSC) usually includes a continuous TiO_2_ layer on Indium Tin Oxide (ITO) or Fluorine-doped Tin Oxide (FTO) glass. This layer, known as the blocking layer (BL), is essential for preventing hole transfer, thereby enhancing the electron conductivity yield of the PSC. Another purpose of the BL is to serve as an electron transfer layer [13]. In our previous works, we introduced a concept of TiO_2_ microdot array with embedded gold nanoparticles (TiO_2_-Au_MDA) placed on the BL to enhance the performance of the PCS. In this method, the BL is deposited using magnetron sputtering, followed by annealing at 550 °C to achieve the anatase phase, which is known to be beneficial for the performance of PSCs. Following the formation of the BL, the MDA structures are first printed as ink on top of the BL and then subjected to a slow heating process, gradually reaching 550 °C for annealing. Namely, the method for creating PSCs with MDA involves a special annealing procedure for the BL, followed by the MDA printing and annealing. At the same time, in both processes, the same temperature is eventually reached. Notably, in contrast to the BL, which requires an annealing temperature of 550 °C to reach the anatase phase, the rationale for annealing the MDA is different. The MDA layer requires a gradual heating up to 550 °C to ensure complete solvent evaporation. Nonetheless, since the maximum temperature remains identical for both layers, it might be possible to combine these processes. The following section will examine the BL and MDA annealing procedures by comparing the single vs. double annealing process.

To determine the optimal BL treatment for maximizing MDA’s impact on PSC performance, we prepared three types of samples with printed TiO_2_-Au_MDA. Type one consisted of samples where both the BL and MDA layers were subjected to separate annealing processes. The second type of dots was printed on a freshly deposited BL without prior annealing, followed by a heat treatment that was applied to both the BL and MDA simultaneously. Lastly, the third type consisted of an annealed BL treated with air plasma before MDA printing and subsequent annealing. Further in the text, we will refer to the first process as BL_A, the second as BL_F, and the third as BL_P. The illustration of PSC with and without MDA is presented in Appendix A.

Figure 1 illustrates the effect of various BL treatments on the diameter of the micro-dots in the TiO_2_-Au_MDA after the annealing process. Although we use the same initial ink volume for each droplet, it spreads differently depending on the BL treatment (Figure 1A–C). The diameter of the micro-dots printed on fresh BL (BL_F) and plasma-treated BL (BL_P) is 90 μm, while on the annealed BL (BL_A), it is 40 μm. This difference results from the fact that an annealed TiO_2_ BL is more hydrophobic compared to the freshly deposited one (see Appendix A for contact angle measurements). The overview of the dots printed on different types of BL is presented in Appendix A. The increment of hydrophobicity is due to the reduction of hydroxyl groups, increased crystallinity, and modified surface morphology achieved through the annealing process [14,15]. The hydrophobic surface has fewer sites that attract water, making it less prone to spreading liquids uniformly. When printing on an annealed layer, this hydrophobic behavior causes the printed droplets to form smaller contact areas or dots with reduced diameters compared to the as-deposited, more hydrophilic layer.

On the other hand, plasma treatment can improve the hydrophilicity of the surface, making it more wettable and increasing surface energy [16]. This improves the wettability for subsequent layer deposition, allowing the micro-dots to spread more freely, resulting in a larger diameter. To summarize, the roughness of the annealed BL makes the surface more hydrophobic, resulting in smaller microdot diameters. In contrast, the fresh and the plasma-treated BL have a hydrophilic surface due to their smoothness, hydroxyl groups, or highly active groups.

This relationship underscores the influence of surface characteristics, such as roughness, on ink-spreading dynamics, as shown in Figure 1D–F, the higher SEM magnifications. On BL_F and BL_P exhibiting lower surface roughness and a higher amount of hydrophilic sides, the ink spreads more extensively across the surface compared to BL_A. This enhanced spreading results in a larger dot diameter and, consequently, a thinner deposited MDA film for the same printed ink volume. The reduced roughness minimizes pinning sites for the ink, allowing it to flow more freely, which negatively impacts the uniformity and distribution of nanoparticles. As the diameter of the MDA increases, its thickness decreases, and the concentration of AuNPs per unit area also diminishes. Thus, the surface modification induced by varying BL treatments offers a means to control the thickness of the MDA layer, enabling precise tuning of nanoparticle distribution and overall layer morphology.

Figure 1G–I shows the Energy-Dispersive X-ray Spectroscopy (EDS) spectra of the analyzed samples. Clear characteristic peaks corresponding to gold (Au) and titanium dioxide (TiO_2_) are observed, confirming their presence in the material. Specifically, the spectrum displays the Au M (~2100 eV), alongside Ti Lα (~400 eV) and O Kα (~500 eV) peaks, which are consistent with TiO_2_. As shown in Figure 1G–I, the EDS peaks for Au and TiO_2_ are more pronounced in the BL_A sample compared to BL_F and BL_P, indicating a higher concentration of both components per unit area in the annealed BL sample. For spatial distribution and elemental mapping, refer to Appendix A.

To summarize, we found that the surface roughness and hydrophobicity of the BL affect the diameter of MDA droplets during printing. This variation in droplet size is expected to influence the optoelectronic properties of the resulting layer, including absorbance and plasmonic behavior—an effect examined in the following section.

As shown in Figure 2A, the absorbance peak for the BL_A sample occurs at 583 nm, while the peaks for the BL_F and BL_P samples shift to longer wavelengths, at 614 nm and 620 nm, respectively. The redshift of the absorption peaks observed in the fresh and plasma-treated BL samples suggests changes in the local dielectric environment and nanoparticle morphology. These effects are likely due to differences in the spreading behavior of the printed ink, which influences the thickness and distribution of AuNPs within the MDA. Surface treatments, such as annealing and plasma exposure, modify the BL’s wettability and roughness and directly affect how the ink settles and dries, ultimately altering the density and aggregation state of the AuNPs—factors that are known to shift plasmonic resonance features. Furthermore, the AuNPs formed within MDA in the annealed BL demonstrate a higher absorbance intensity and a narrower full width at half maximum (FWHM). Here, the FWHM is 445 nm, 582 nm, and 536 nm for BL_A, BL_F, and BL_P, respectively. A lower FWHM indicates a more uniform size distribution of AuNPs, whereas broader peaks in the BL_F and BL_P samples point to increased polydispersity or aggregation of nanoparticles [17]. This difference highlights the impact of annealing in promoting more uniform nanoparticle growth and deposition, potentially through enhanced nucleation and stabilization processes on the BL surface. These findings suggest that annealing the BL provides better control over AuNP size and distribution, resulting in improved optical properties. In contrast, lower surface roughness in fresh and plasma-treated BLs may allow nanoparticles to spread unevenly, leading to wider particle size distribution and reduced absorption efficiency. This observation underscores the importance of tailoring BL treatment to optimize nanoparticle morphology and enhance device performance. Future studies could explore the correlation between these optical properties and the photovoltaic performance of the corresponding devices.

The absorbance measurements indicate that the strongest localized surface plasmon resonance (LSPR) contribution is observed in the BL_A sample with a ~40-micron diameter MDA. A smaller MDA diameter increases the concentration of Au NPs per area, leading to a higher absorption peak driven by an enhanced plasmonic effect. However, further examination of differently treated BL layers and their corresponding LSPR contributions is necessary.

Figure 2B–F and Table 1 compare the J-V performance of devices fabricated using different types of surface treatment of BL. According to Table 1, the best performance is observed in the two-step annealing treatment, specifically, the BL-A and TiO_2_-Au_MDA PSC, which has a micro-dot diameter of 40 µm. However, in Table 1, devices demonstrating the highest PCE do not necessarily exhibit the highest value for each individual parameter, such as FF or Jsc [18]. Perovskite layers fabricated by spin-coating often contain defects and structural heterogeneity, leading to nonradiative recombination that reduces V_OC_. However, V_OC_ losses in PSCs remain relatively small compared to other solution-processed photovoltaics. In our experiments, the V_OC_ variation within sample groups was similar to that between different processing groups, suggesting that preparation variability and BL treatment have a stronger influence on defect formation, particularly at interfaces with the perovskite and hole transport layers [19]. Similarly, the statistical analysis of the J-V performance (Figure 2C,E) reveals that the sample with BL_A also exhibits the highest values for Jsc and PCE. Performance data for the standard PSC (without MDA) and for TiO_2_-MDA without AuNPs are provided in the Appendix A. The enhanced performance of the samples with BL_A is induced by repeating the annealing procedure on BL. This can allow for further growth of crystalline grains, promoting a more stable and ordered anatase structure [20]. Notably, the anatase faces of TiO_2_ offer better electronic properties, such as a higher conduction band edge and lower electron recombination rates, compared to rutile and brookite [21]. In addition, each annealing step reduces defects, improves grain boundary coherence, and enhances phase purity. This leads to better electron mobility and conductivity in the anatase TiO_2_ layer, which is beneficial in applications like perovskite solar cells. Different treatments of the TiO_2_ BL provide a method to control the thickness of the TiO_2_-Au_MDA layer, enabling precise tuning of the gold nanoparticle distribution and its optical absorption peak. The plasmonic effect is most pronounced with the BL_A, highlighting that the two-step annealing process is the most effective in enhancing the performance and impact on the PSC. The improved solar cell performance of the PSC with BL_A strongly suggests that this is the preferred approach for preparing AuNP-enhanced solar cells.

Understanding the optimal size and location of AuNPs within the BL (TiO_2_) layer in PSCs is essential for optimizing plasmonic enhancement effects. That is to say, the LSPR induced by AuNPs can improve light absorption, but its effectiveness depends on nanoparticle size and distribution. In order to investigate these parameters systematically, FDTD simulations were performed using Lumerical software. The goal here is to determine the optimal AuNP size that maximizes absorption enhancement while remaining compatible with the physical constraints of the device architecture.

The results of the simulation are presented in Figure 3A. The curves show the negative transmittance (−T) as a function of wavelength for cells incorporating AuNPs with diameters of 10, 20, 30, and 40 nm alongside a reference cell without nanoparticles (“Clear Cell”) for comparison. The schematic cells are presented in Figure 3B,C, where the size of the AuNPs has changed. Outside the solar cell, the transmittance monitor was positioned below the entire device stack, so the signal represents the total light absorbed in all layers. The main observation from this comparison is that increasing the size of the AuNPs leads to a stronger plasmonic response, suggesting a more pronounced impact on PSC performance with larger particles. This result aligns well with previous reports on the size-dependent behavior of AuNPs, where larger particles exhibit enhanced LSPR features [22]. Indeed, at small diameters (10 nm and 20 nm), the AuNPs do not exhibit a distinct absorption peak, indicating weaker plasmonic activity in this size range. A more pronounced enhancement is observed for 30 nm and especially 40 nm AuNPs, where distinct absorption peaks emerge around 680–720 nm. Notably, the 40 nm AuNPs lead to the most significant plasmonic contribution, evidenced by the broad absorption shoulder in the red region. These findings confirm that both the presence and the size of AuNPs play a critical role in modulating light absorption, with optimal enhancement achieved in the size range of 30–40 nm. The size-dependent enhancement trend is observed in the absorbance spectrum. However, at a size of 30 nm, a noticeable absorption peak emerges, signifying the onset of significant plasmonic enhancement. While the trend suggests that increasing AuNP size enhances plasmonic effects, experimental limitations must be considered. Larger nanoparticles often exhibit greater size deviations during synthesis, leading to a broader resonance spectrum and reduced enhancement due to non-uniform plasmonic coupling. Additionally, since the thickness of the TiO_2_ layer is constrained to 40 nm, larger AuNPs would necessitate a thicker BL, potentially increasing charge transport resistance, working as recombination spots, and negatively affecting PSC efficiency. Hence, choosing a 30 nm diameter for AuNPs represents a promising balance between optical gain and structural feasibility.

The position of AuNP-containing printed layers significantly influences the PSC performance. In previous experimental results, printing AuNP-containing dots on top of the BL demonstrated superior current density Jsc compared to configurations where the dots were printed below the BL. To further understand the underlying mechanisms driving these performance differences, we conducted FDTD simulations of both architectures. By comparing simulated and experimental results, we aimed to gain deeper insights into how nanoparticle placement affects charge transport and light absorption in PSCs.

Following the understanding of the optimal BL treatment, further investigation will focus on evaluating the structural arrangement of the layers within the PSC. The positioning of the MDA is anticipated to play a critical role in determining its interactions with adjacent layers, directly impacting key device properties such as recombination resistance, capacitance, and charge transport dynamics. More specifically, concerns remain regarding the direct contact between AuNPs and MAPI, which may enhance electron recombination and negatively impact efficiency [23]. Thus, we examined two architectures of the PSC with MDA. The first architecture involves applying the MDA directly onto the FTO-coated glass, followed by the deposition of the BL. In contrast, the second architecture consists of placing the BL on the FTO-coated glass, followed by the deposition of the MDA. The rationale for choosing these two architectures is twofold. On the one hand, having the MDA covered by the BL may help prevent direct contact with the MAPI layer and reduce undesirable interactions. On the other hand, if AuNPs are positioned too far from the MAPI layer in a PSC, their capacity to enhance performance through LSPR diminishes. The setup of the simulation is shown in Figure 3B,C. Here, we consider two main architectures for simulation: AuNPs located close to the (A) BL/MAPI interface and (B) FTO/BL interface (BL → TiO_2_-Au_MDA is Figure 2A, and TiO_2_-Au_MDA → BL is Figure 2B. Notably, the thicknesses of the layers were chosen using the practical constraints of the “real world” device.

Figure 4 presents the statistical distribution of PSC performances and simulated Jsc values. From statistical analysis, both the distribution and the overall results show a more significant benefit to the BL → TiO_2_-Au_MDA structural order. In the sample with TiO_2_-Au_MDA → BL, the Au NPs were placed inside TiO_2_, with the thickness of the MDA 40 nm [9], having an additional 40 nm of BL coating. Indeed, the results presented in Figure 4 show that this distance is too far from MAPI to benefit PSC. This occurs because the electromagnetic field generated by the AuNPs weakens with distance, reducing light absorption enhancement and limiting the impact on carrier generation [24]. As a result, the plasmonic effect becomes less effective, leading to suboptimal improvements in efficiency. Proper proximity is essential to balance optical enhancement and prevent charge recombination. Moreover, the AuNPs come into direct contact with the FTO layer in the second case, potentially creating conductive pathways that bypass the intended charge transport layers. This leads to unintended electrical connections, resulting in short circuits that diminish the overall efficiency and performance of the solar cell. To summarize this part, we found that the optimal architecture for the most beneficial performance of PSCs is BL → TiO_2_-Au_MDA.

The simulation results presented in Figure 4 Jsc strongly correlate with experimental findings, confirming that nanoparticle positioning significantly affects the efficiency of PSCs. When AuNP-containing microdots were positioned at the BL/MAPI interface, the simulated short-circuit current density Jsc was 15.7 mA/cm^2^, closely aligning with the experimentally observed average of 18.7 mA/cm^2^. In contrast, when the dots were placed at the FTO/BL interface, the simulated Jsc was 7.9 mA/cm^2^, which also matched well with the experimental average of 6.9 mA/cm^2^. The significant difference in performance between the two AuNP positions can be attributed to both optical and electronic effects associated with plasmonic enhancement near the active layer. When AuNPs are located at the BL/MAPI interface, they reside in close proximity to the photoactive perovskite layer, allowing efficient coupling of the LSPR field to the perovskite absorber. This proximity enhances the near-field intensity and increases light absorption, specifically within the region where exciton generation occurs, thereby directly boosting photocurrent. In contrast, AuNPs placed near the FTO/BL interface are spatially separated from the absorber, and the plasmonic near-field decays before effectively interacting with the perovskite layer, leading to diminished enhancement. Additionally, embedding AuNPs too close to the transparent conducting oxide (FTO) can introduce parasitic absorption and increased recombination at the interface, negatively impacting charge extraction and overall device efficiency [2,3,25]. Thus, positioning AuNPs at the BL/MAPI interface offers an optimal balance between plasmonic light trapping and efficient charge collection. This observation aligns with previous studies emphasizing the importance of nanoparticle placement relative to the absorber layer in plasmon-enhanced perovskite solar cells. The close agreement between simulated and experimental Jsc values validates the underlying mechanisms and highlights the practical significance of spatial nanoparticle engineering for performance optimization in next-generation solar cells.

PSC performance was re-evaluated after seven days of storage to evaluate the stability of the devices. Table 2 shows the stability measurements of PSCs with different MDA architectures over seven days, revealing clear trends in device performance retention. For the architecture where the BL is deposited first and the TiO_2_-Au_MDA is printed atop (BL → TiO_2_-Au_MDA), the photovoltaic parameters remain stable, with Jsc slightly increasing from 18.6 to 19 mA/cm^2^ and PCE showing minimal decrease from 3.5% to 3.3%. This suggests that placing the AuNP-containing MDA closer to the perovskite layer not only enhances initial performance but also preserves device stability over time, potentially due to improved interface contact and minimized degradation pathways.

In contrast, the reverse architecture (TiO_2_-Au_MDA → BL) exhibits both lower initial performance and notable degradation upon storage. The PCE decreases from 1.7% to 0.9% within seven days, with a corresponding drop in Voc and FF, indicating increased recombination losses and interface instability. The consistently low Jsc values in this configuration suggest limited plasmonic enhancement due to the increased distance of AuNPs from the absorber layer and potential detrimental interactions with the underlying FTO or BL layers. Full results with the standard device and TiO2_MDA are presented in Appendix A. Overall, these results demonstrate that the spatial positioning of the AuNP-containing MDA critically affects not only the immediate performance but also the operational stability of PSCs, underscoring the necessity of precise interfacial engineering for achieving durable plasmonic-enhanced devices.

Following the MDA positioning on the top or under the BL results, we examined three AuNP concentrations in MDA to evaluate the influence of concentration on the stability of the precursor dispersion and on device performance. The concentration of AuNP in the TiO_2_ matrix plays a crucial role in determining the overall efficiency and stability of the resulting solar cell. If the AuNP concentration is too high, it can lead to excessive light scattering, increased recombination sites, and reduced transparency, which may hinder the effective absorption of incident light. High densities of AuNPs may also lead to particle agglomeration, compromising light-harvesting efficiency and counteracting the benefits of plasmonic enhancement [26]. If the AuNP concentration is too low, plasmonic enhancement becomes insufficient, reducing LSPR effects, light absorption, and the benefits of field enhancement and scattering. This leads to only marginal PCE improvements compared to cells without nanoparticles [23]. Therefore, an optimal concentration of AuNPs is essential. It must be high enough to induce significant plasmonic effects and enhance light absorption while low enough to prevent aggregation, maintain transparency, and minimize recombination losses. Thus, optimal concentration is essential to balance enhancement and reduce losses, maximizing solar cell efficiency.

To determine the optimal concentration of AuNPs, three different molar ratios of HAuCl_4_ (0.017 M, 0.025 M and 0.034 M) were used in the ink. From here onward, we will mark the inks and printed samples of 100%-0.017 M, 150%-0.025 M, and 200%-0.034 M as TiO_2_-1×Au_MDA, TiO_2_-1.5×Au_MDA, and TiO_2_-2×Au_MDA, respectively. Notably, the Au ions, tightly bound to the TiO_2_ precursor, form nanoagglomerates that exhibit stable colloidal behavior within the ink [9]. Therefore, analyzing the inks with various Au concentrations will shed light on the future properties of the PSC. The molar concentration (0.017 M (0.33 wt% Au ions in the ink), 0.025 M (0.49 wt% Au ions in the ink), and 0.034 M (0.65 wt% Au ions in the ink)) refers to the HAuCl_4_ precursor in the ink formulation prior to printing. In contrast, the nominal 23 wt%, 34.5 wt%, and 46 wt% represent the initial gold content relative to TiO_2_ in the printed microdot array after annealing. These two values describe different stages of material preparation and are both provided for completeness.

The prepared inks of TiO_2_-×Au were analyzed using dynamic light scattering (DLS) and zeta potential (Z-Pot) techniques (Figure 5). We find that the mean hydrodynamic diameter (HDD) of TiO_2_-Au complex nanoparticles in ink, presented in Figure 5A, was about 120 nm with a relatively narrow size distribution (FWHM is 140, 160, 210 nm for 100%, 150%, and 200% TiO_2_-Au. Notably, as the concentration of the Au precursor increases, both the HDD and the corresponding FWHM also exhibit an increase. Z-potential (Z-Pot) is a parameter that can be used to quantify the charge stability of the solution and, thus, the colloidal stability of the ink. We found that across all Au concentrations, the Z-Pot value was approximately 30 mV (Figure 5B), indicating the ink’s stability [27]. Notably, higher concentrations of Au ions result in increased Z-Pot values. An absolute Z-Pot value above 60 mV signifies excellent stability, suggesting that higher concentrations yield more stable colloids [28].

Increasing the concentration of AuNP precursor in ink not only enlarges the particle size in the MDA layer but also enhances LSPR intensity, potentially altering the absorption spectrum and improving light-harvesting capabilities. However, excessive concentration may cause NPs agglomeration, compromising film uniformity and overall device performance.

While all tested concentrations of AuNPs demonstrated stable suspensions suitable for inkjet printing, further investigations are essential to determine the optimal formulation for achieving uniform deposition and consistent performance. Thus, a systematic series of experiments was conducted to evaluate the impact of AuNP concentration on the diameter of MDAs, nanoparticle size, and plasmonic absorbance. Based on the results presented above, a 40 µm diameter of TiO_2_-Au_MDAs was chosen for use in solar cell fabrication. For SEM analysis, samples were prepared by printing MDA with different concentrations of gold on FTO substrates. Additionally, MDA layers with the same dot diameters were printed on glass substrates to measure absorption.

The TEM micrographs of the TiO_2_-×Au_MDA on the TiO_2_ BL after annealing are presented in Figure 6A–C and illustrate the presence and arrangement of AuNPs in a section matrix. The average size of AuNPs was calculated using “ImageJ 1.54g” software, and Figure 6D–F presents the size distribution of AuNPs. The AuNP size distributions varied notably with precursor concentration. For TiO_2_-1×Au_MDA, three distinct size populations were identified: a small subset (~10%) with sizes in the range of 8 ± 3 nm, a dominant group (~50%) centered at 30 ± 5 nm, and a significant fraction (~40%) consisting of larger particles around 45 ± 6 nm. TiO_2_-1.5×Au_MDA exhibited a more uniform bimodal distribution, comprising a smaller population (~15%) at 7 ± 3 nm and a dominant group (~75%) at 30 ± 6 nm, indicating improved homogeneity relative to TiO_2_-1×Au_MDA. In contrast, TiO_2_-2×Au_MDA showed a broader size distribution, with two main populations (~40% and ~60%, respectively) at 17 ± 6 nm and 51 ± 5 nm, suggesting pronounced particle growth and aggregation at higher precursor concentrations. Moreover, the calculated D50 values based on AuNP size distribution analysis are 35 nm for the 1× concentration, 30 nm for the 1.5× concentration, and 48 nm for the 2× concentration. The notable increase in D50 for the 2× sample indicates a shift toward larger particle sizes, supporting the conclusion that higher precursor concentrations promote increased aggregation or coalescence of nanoparticles.

The similarity in nanoparticle sizes between TiO_2_-1×Au_MDA and TiO_2_-1.5×Au_MDA indicates that both compositions of ink maintain controlled nucleation and growth conditions. However, the absence of a third size group in TiO_2_-1.5×Au_MDA suggests a more uniform particle distribution, which could enhance plasmonic effects while minimizing potential defects or scattering losses [29]. In contrast, the substantial size increase in TiO_2_-2×Au_MDA raises concerns about its compatibility with the TiO_2_ matrix. The presence of AuNPs exceeding the thickness of the TiO_2_ layer increases the likelihood of direct contact with the MAPI absorber, which could introduce unwanted charge recombination pathways, potentially hindering PSC performance. This phenomenon is particularly evident in Figure 6C for the TiO_2_-2×Au_MDA sample, where larger nanoparticles are present among the predominantly smaller ones. Moreover, the particle size analysis suggests that higher precursor concentrations may facilitate multiple nucleation events, where new nuclei form instead of allowing existing particles to grow larger, ultimately reducing average particle size [30]. Namely, increased Au precursor concentration leads to faster reduction kinetics, where the rapid formation of nuclei limits diffusion-controlled growth, ultimately yielding smaller particles [31]. The excess precursor can also induce localized aggregation, altering growth dynamics and resulting in non-uniform particle formation [32]. Therefore, TiO_2_-1.5×Au_MDA appears to be the optimal concentration, balancing size uniformity, LSPR enhancement, and compatibility within the PSC structure. Nevertheless, the NPs’ density per unit volume within the TiO_2_ microdot is notably higher in TiO_2_-1.5×Au_MDA compared to TiO_2_-1×Au_MDA, which may increase the likelihood of “bare” AuNPs coming into direct contact with the MAPI layer. Such direct interfaces can act as recombination centers, ultimately reducing solar cell performance.

To understand how these AuNP size and distribution variations influence the optical response, absorbance measurements were conducted across the different TiO_2_-Au_MDA samples (Figure 6G–I). Notably, AuNPs in the 10–35 nm size range typically exhibit a resonance around 520 nm [33]. However, when incorporated into a medium with a higher refractive index, such as a TiO_2_ matrix, their plasmonic peak shifts toward longer wavelengths [34]. Indeed, for TiO_2_-1× and 1.5×Au_MDA, the primary LSPR peak is observed at 542 nm with an FWHM of 41 and 59 nm, respectively. Additionally, a smaller peak appears at 589 nm and 566 nm, with a broader FWHM of 56 and 134 nm for TiO_2_-1× and 1.5Au_MDA, respectively. The increase in FWHM further supports the broader size distribution observed in the 1.5×AuNP material, as revealed by TEM micrograph analysis, compared to the more uniform distribution in the 1×AuNP counterpart.

Moreover, the maximum of the secondary peak at the 1× concentration shifts toward the infrared region, likely due to the combined plasmonic response of the larger AuNP population (~45 nm). This redshift, attributed to the presence of these larger particles, is less pronounced at the 1.5× concentration, where a more uniform size distribution limits the contribution of multiple plasmonic modes. For TiO_2_-2×Au_MDA, the LSPR peak shifts to 571 nm with a markedly broader FWHM of 110 nm, accompanied by an additional peak at 608 nm with an FWHM of 145 nm. These spectral features correlate with the larger average nanoparticle size observed in the TEM micrographs (Figure 6C), as increased particle dimensions typically result in resonance at longer wavelengths [35,36]. The increased FWHM values further confirm a highly polydisperse system with a broad range of particle sizes, contributing to excessive light scattering and potential plasmonic damping. Such effects may reduce the uniformity of the plasmonic enhancement, potentially limiting light absorption. This, in turn, could negatively impact the photovoltaic performance of the PSC by introducing recombination sites and reducing the efficiency of charge carrier generation and collection [37]. Overall, the observed LSPR trends support the nanoparticle size distribution findings, where increasing AuNP concentration leads to broader FWHM values and peak shifts, reflecting greater polydispersity.

To better understand how the presence of AuNPs affects the structure of the perovskite films, MAPI layers were deposited on glass substrates with and without inkjet-printed TiO_2_-×Au_MDA, prepared with different gold precursor concentrations (1×, 1.5× and 2×), corresponding to the samples MAPI_TiO_2_-1×Au_MDA, MAPI_TiO_2_-1.5×Au_MDA and MAPI_TiO_2_-2×Au_MDA, respectively.

From the comprehensive lattice parameter refinement analysis (for full phase analysis, see Appendix A XRD) of the diffraction patterns presented in Figure 7A, it is evident that all samples are polycrystalline, single-phase, and homogeneous. Their crystallographic structure corresponds to the symmetry described by the I4/mcm space group [38,39]. Importantly, no reflections corresponding to lead iodide (PbI_2_) were observed in any of the samples, indicating the phase purity of the MAPI films and the absence of detectable degradation or secondary phase formation. This demonstrates that the incorporation of TiO_2_-×Au_MDA at the interface and variation in AuNP concentration do not disrupt the crystallographic structure or homogeneity of the perovskite layer [40]. Additionally, the diffraction patterns show no visible contributions from the underlying TiO_2_ matrix, gold nanoparticles, or the amorphous glass substrate. This is attributed to the grazing-incidence measurement geometry (incident angle of 0.5°), which significantly limits X-ray penetration depth and enhances surface sensitivity. As a result, the signal is dominated by the upper MAPI film, while contributions from deeper, non-crystalline, or low-scattering layers remain undetected.

To further evaluate the internal microstructure of the MAPI films, the average crystallite sizes (coherent scattering domain sizes) were extracted from the XRD peak broadening using the Halder–Wagner method (Figure 7B) [41]. The analysis revealed that the incorporation of the TiO_2_-Au_MDA at the BL/MAPI interface results in an increase in crystallite size by approximately 17% compared to the reference MAPI film deposited on a bare glass substrate. Importantly, no statistically significant variation in crystallite size was observed among the films containing different concentrations of Au precursor (1×, 1.5× and 2×), indicating that the crystallite size is not directly affected by the concentration of AuNPs. Instead, the observed enhancement in crystallite size is attributed primarily to the altered substrate morphology caused by the printed square array structure, which modifies the nucleation and growth dynamics of the perovskite during film formation. Since all three modified substrates share the same printed geometry (40 µm square arrays), their influence on MAPI crystallization is effectively equivalent, regardless of the actual amount or distribution of AuNPs (Appendix A).

Figure 7C presents the dependence of the MAPI unit cell volume on the Au concentration in the MAPI_TiO_2_-×Au_MDA samples. Data analysis shows that the MAPI unit cell volume variation does not exceed 0.5%. Within experimental error, the unit cell volume remains essentially independent in all samples, including modified ones. This lack of a direct structural impact from the embedded AuNPs indicates the potential of the proposed indirect modulation approach, whereby the optoelectronic properties of the MAPI film can be tuned via the incorporation of an ordered array of active sites without compromising the crystal structure. Such behavior suggests that AuNPs do not chemically integrate into the perovskite structure but instead reside beneath the active layer, acting as external stimuli that modify local crystallization dynamics without inducing lattice distortion. Importantly, this finding demonstrates that the proposed indirect modulation strategy enables a twofold increase in AuNP precursor concentration relative to the initial loading without compromising key structural parameters of the perovskite. Since structural integrity is critical for charge generation and transport in perovskite solar cells, the ability to vary AuNP content over a broad range while preserving the crystal quality of MAPI highlights the robustness and scalability of this approach for device engineering.

The structural analysis confirms that incorporating TiO_2_-×Au_MDA arrays at the BL/MAPI interface does not disturb the crystallographic phase or integrity of the MAPI films across all investigated AuNP concentrations. The crystallite sizes remain largely unaffected within the range of experimental uncertainty, and no secondary phases or structural distortions are observed. These findings highlight a key advantage of the proposed interfacial design: it enables the integration of plasmonic nanoparticles in close proximity to the perovskite layer without embedding them directly into the active material. This strategy preserves the structural stability of the light-absorbing layer, a critical requirement for maintaining device performance while opening new opportunities to enhance optoelectronic properties through field-enhanced or morphological interactions. Moreover, this interfacial approach offers a broad design space for tuning nanoparticle type, concentration, and spatial arrangement without compromising the perovskite’s microstructure.

To correlate the observed variations in nanoparticle size, LSPR behavior, and interaction with the MAPI layer to actual device performance, the next step was to assess the impact of varying AuNP concentrations on PSC operation. A series of diode-like devices was fabricated to detect and analyze these effects, maintaining the same layer architecture as the full PSC but excluding the MAPI layer. This approach allows for a more precise evaluation of the junction properties and the intrinsic rectification behavior of the TiO_2_-HTL interface, independent of the photovoltaic effect introduced by the perovskite absorber. Namely, by investigating these simplified structures, it becomes possible to directly assess how different AuNP concentrations influence charge transport, barrier formation, and carrier recombination at the BL and HTL interface. The samples with various concentrations of AuNP were made while maintaining a consistent device structure: FTO–BL TiO_2_–TiO_2_-×Au_MDA–HTL–electrode (Appendix A).

Figure 7D shows the electrical performance of the diodes with different AuNP concentrations. The curves were analyzed by extracting resistance values from the linear region of the dark J-V curves. The resistance values for the different sample configurations indicate significant variations in charge transport behavior depending on the concentration of the TiO_2_ NP. The measured resistance values were 2.3 ± 0.1 Ω·cm^2^ for TiO_2_-1×Au_MDA, 6.2 ± 0.1 Ω·cm^2^ for TiO_2_-1.5×Au_MDA and 6.0 ± 0.1 Ω·cm^2^ for TiO_2_-2×Au_MDA. These results demonstrate a clear trend where increased AuNP concentration leads to higher resistance, negatively affecting charge transport. Despite the broader size distribution observed in the TiO_2_-1×Au_MDA sample, including a notable population of larger nanoparticles (~45 nm), this sample exhibited the best diode performance. This suggests that at 1× concentration, the AuNP content and distribution strike an optimal balance between plasmonic enhancement and charge transport. Although the 1.5× sample showed more uniform particle sizes, its performance was slightly lower, indicating that size uniformity alone is insufficient to improve device behavior. Notably, the resistance of the 1.5× and 2× samples was nearly identical (~6.0 Ω·cm^2^), suggesting a saturation effect and indicating that beyond a certain concentration, the drawbacks of excessive AuNP loading—such as increased carrier scattering, percolation blockage, and interfacial recombination—dominate the system behavior [42,43].

To summarize, moderate AuNP loading (1×) appears optimal: it achieves a balanced size distribution, leverages the LSPR enhancement of ~30–45 nm particles, and maintains favorable charge transport [44]. In contrast, higher loadings (1.5× and 2×) introduce excessive particle densities that disrupt film morphology and increase resistance [45,46]. Following the diode-like device characterization, which examined the impact of varying AuNP concentrations on charge transport, the next step was to evaluate the full photovoltaic performance of the corresponding structures. To this end, PSCs incorporating TiO_2_-×Au_MDA atop the BL were fabricated by introducing the MAPI perovskite absorber layer. All devices shared the same architecture shown in Appendix A: FTO/BL/TiO_2_-×Au_MDA/MAPI/HTL/Electrode, and, as previously, were prepared with three Au_MDA concentrations: 1×, 1.5×, and 2×.

Key performance parameters, including short circuit current density (Jsc), open circuit voltage (Voc), fill factor (FF), and power conversion efficiency (PCE), were extracted from JV measurements and are presented in Figure 8 and Appendix A. The results revealed a decreasing trend across all metrics with increasing AuNP content. The TiO_2_-1×Au_MDA sample exhibited the best performance, with Jsc = 7 ± 2 mA/cm^2^, Voc = 0.74 ± 0.02 V, FF = 30 ± 10% and PCE = 1.7 ± 0.2%. Moderate loading (1.5×) yielded slightly lower values (Jsc = 6 ± 1 mA/cm^2^, Voc = 0.73 ± 0.01 V, FF = 36 ± 10%, PCE = 1.6 ± 0.4%), while the highest loading (2×) resulted in a most pronounced decline: Jsc = 4 ± 1 mA/cm^2^, Voc = 0.67 ± 0.03 V and PCE = 0.5 ± 0.2%. These photovoltaic results align with the diode-like measurements, confirming that excessive AuNP incorporation detrimentally affects device performance. The optimal performance of the 1× sample can be attributed to balanced plasmonic enhancement provided by ~30–45 nm AuNPs without significantly perturbing the TiO_2_ matrix [2,3,4,5,6]. As evidenced by low resistance and relatively high Jsc and Voc values, this configuration supports efficient charge transport and reduced recombination [47]. In contrast, the 1.5× and 2× samples exhibit increased resistance and morphological inhomogeneity due to higher nanoparticle densities and broader size distributions. While moderate AuNP incorporation can enhance light harvesting via LSPR [3,48], as seen in the redshifted absorption spectra (Figure 6G–I), excessive loading disrupts the BL’s electronic properties and interface stability. These disruptions likely introduce charge trapping and recombination centers, leading to reduced Voc, Jsc and FF [49]. The pronounced performance decline in the TiO_2_-2×Au_MDA sample suggests that beyond a certain threshold, the detrimental effects of matrix disruption, increased recombination, and hindered charge extraction outweigh the optical benefits of plasmonic enhancement, ultimately lowering the overall PCE.

## 4. Conclusions

This study presents a comprehensive and systematic exploration of AuNP incorporation into PSCs via an innovative inkjet-printed TiO_2_-AuNP microdot array (MDA) strategy. Unlike previous investigations, this work integrates the interplay among substrate conditioning, spatial nanoparticle arrangement, and AuNP concentration, thereby identifying critical parameters significantly influencing photovoltaic performance. Specifically, we determined that annealing the TiO_2_ BL markedly enhances surface hydrophobicity, producing smaller, denser microdots with pronounced LSPR. Further, careful placement of the MDA at the interface between the BL and the perovskite absorber, rather than beneath the BL, substantially elevates device efficiency due to enhanced near-field coupling and reduced carrier recombination, effects thoroughly supported by FDTD simulations. Crucially, the structural compatibility of this novel plasmonic interface with the MAPI layer is confirmed via XRD analysis, which reveals no detectable structural distortions or secondary phase formations. Although higher AuNP concentrations improve dispersion stability, maintain MAPI crystallographic integrity, and yield more uniform nanoparticle size distributions, diode-like and photovoltaic measurements demonstrated that these advantages do not translate into better device performance. The 1× (~23 wt% after annealing) concentration provides optimal PSC efficiency by achieving the best balance between plasmonic enhancement and charge transport while avoiding the increased resistance, carrier scattering, and recombination losses observed at higher loadings.

Thus, this systematic investigation not only underscores the efficacy and practicality of employing inkjet-printed plasmonic microarrays but also provides unique insights and clear guidelines for future interface engineering, significantly advancing the optimization and scalability of next-generation perovskite photovoltaics.

## Figures and Tables

**Figure 1 nanomaterials-15-01501-f001:**
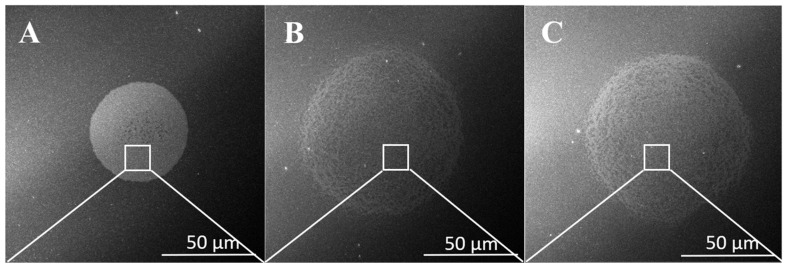
SEM micrographs showing printed TiO_2_-Au_MDA microdots on TiO_2_ blocking layers (BL) with different surface treatments: (**A**) annealed BL (BL_A), (**B**) freshly deposited BL (BL_F), and (**C**) plasma-treated annealed BL (BL_P). The dot diameter is ~40 μm on BL_A and ~90 μm on both BL_F and BL_P, demonstrating the influence of surface treatment on ink-spreading behavior. Higher SEM magnification of the MDA is presented in: (**D**) annealed BL (BL_A), (**E**) freshly deposited BL (BL_F), and (**F**) plasma-treated annealed BL (BL_P). EDS is presented in figures: (**G**) annealed BL (BL_A), (**H**) freshly deposited BL (BL_F), and (**I**) plasma-treated annealed BL (BL_P).

**Figure 2 nanomaterials-15-01501-f002:**
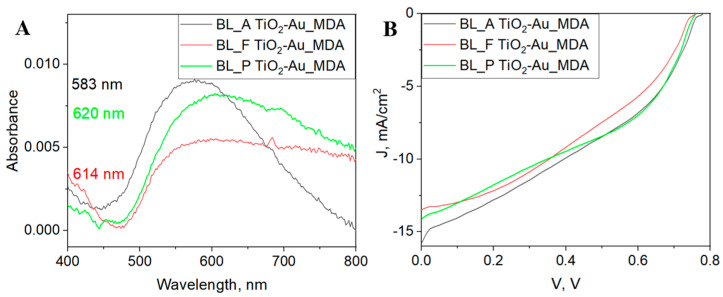
(**A**) Absorbance spectra of TiO_2_ BL with different surface treatments: BL_A (black curve), BL_F (red curve), and BL_P (green curve). Note that the absorbance was derived using the reflectance measurements of the films. (**B**) J–V characteristics of the PSCs. (**C**–**F**) Box plots summarizing the photovoltaic parameters extracted from the J–V curves, including (**C**) short-circuit current density Jsc, (**D**) open-circuit voltage Voc, (**E**) power conversion efficiency PCE, and (**F**) fill factor FF. Performance data for the standard PSC (without MDA) and for TiO_2_-MDA without AuNPs are provided in the Appendix A.

**Figure 3 nanomaterials-15-01501-f003:**
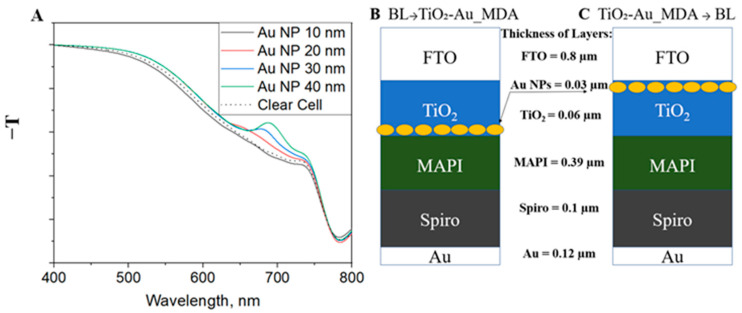
(**A**) Simulated negative transmittance (−T) spectra of perovskite solar cells (PSCs) incorporating AuNPs of varying diameters (10, 20, 30, and 40 nm) are presented alongside a reference device without nanoparticles (“Clear cell”). The transmittance monitor was placed beneath the complete device stack to capture the total light absorption across all layers. (**B**,**C**)—Schematic representations of the two simulated device architectures incorporating AuNPs: (**B**) AuNPs located at the BL/MAPI interface (BL → TiO_2_-Au_MDA), and (**C**) AuNPs positioned at the FTO/BL interface (TiO_2_-Au_MDA → BL).

**Figure 4 nanomaterials-15-01501-f004:**
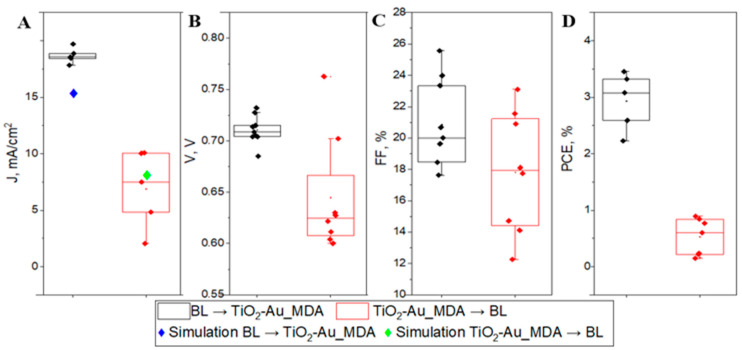
The comparison of PSC performance with MDA positioned below (black) or above (red) BL layer is illustrated by the black and red data boxes, respectively. (**A**) presents short-circuit current density Jsc where the samples are denoted as BL → TiO_2_-Au_MDA (black), TiO_2_-Au_MDA → BL (red), simulations of the device with AuNPs located at the BL/MAPI interface (BL → TiO_2_-Au_MDA blue), and AuNPs positioned at the FTO/BL interface (TiO_2_-Au_MDA → BL green). (**B**) open-circuit voltage Voc. (**C**) power conversion efficiency PCE. (**D**) fill factor FF. Box plots, including standard and TiO_2_ MDA, are presented in Appendix A.

**Figure 5 nanomaterials-15-01501-f005:**
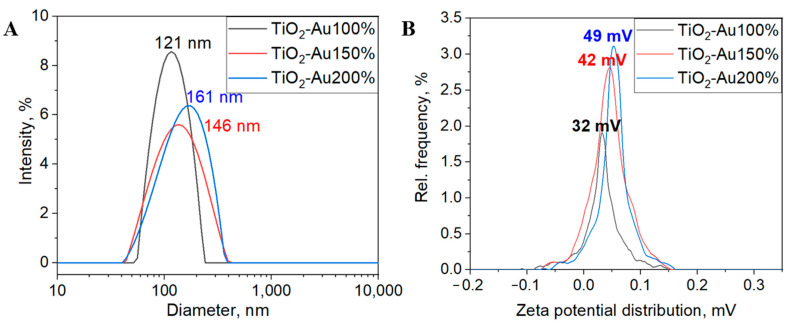
(**A**) Dynamic light scattering (DLS) analysis of TiO_2_/Au inks showing hydrodynamic diameter distributions at different concentrations of Au precursor; (**B**) Z-Pot measurements indicating colloidal stability of the inks.

**Figure 6 nanomaterials-15-01501-f006:**
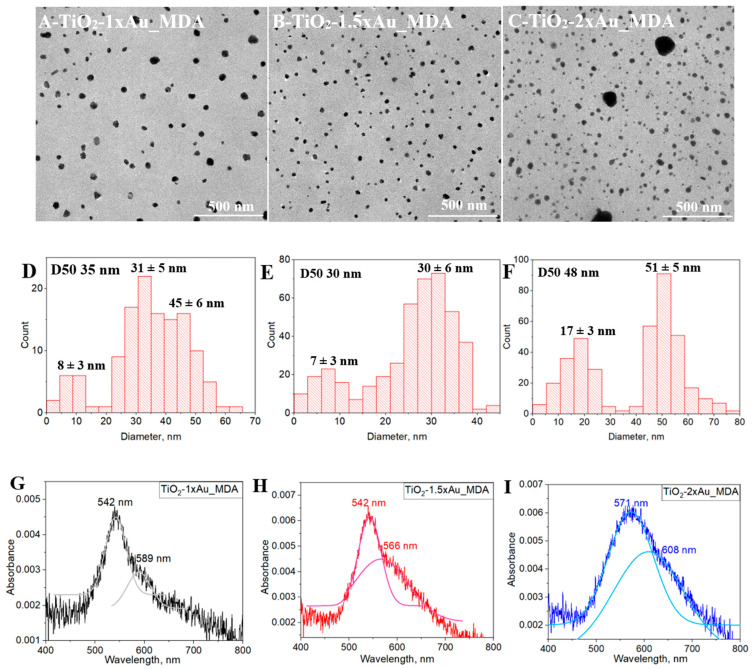
(**A**–**C**) TEM micrographs of TiO_2_-×Au_MDA samples with Au NP precursor concentrations of 1×, 1.5× and 2×, respectively. (**D**–**F**) Corresponding nanoparticle size distributions extracted from TEM images using ImageJ software. (**G**–**I**) Absorbance spectra derived from reflectance measurements of the same samples deposited on glass substrates highlight the effect of AuNP concentration on the plasmonic response.

**Figure 7 nanomaterials-15-01501-f007:**
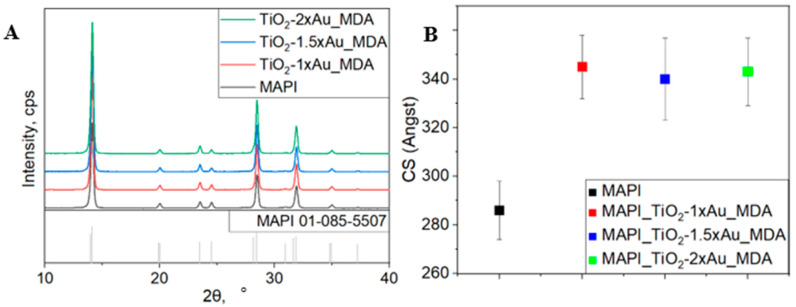
(**A**) X-ray diffraction (XRD) patterns of MAPI films deposited on bare glass substrates (MAPI) and on glass substrates printed with TiO_2_-×Au_MDA layers containing different Au precursor concentrations (x = 1×, 1.5×, 2×), the grey bars represent the known XRD patterns of the cubic phase of MAPI (ICDD 01-085-5507). (**B**) Crystallite sizes (coherent scattering domain sizes) were extracted from the FWHM of selected XRD peaks using the Halder–Wagner method. (**C**) Lattice parameters of MAPI films synthesized on TiO_2_-×Au_MDA substrates with varying Au precursor concentrations. Parameters were obtained from comprehensive XRD refinement with angular correction using a peak shift model function. (**D**) Dark I–V characteristics of diode-like devices incorporating different AuNP concentrations in the TiO_2_-×Au_MDA layer. Increased series resistance is observed with higher AuNP loading, extracted from the slope of the linear region in the J–V curves.

**Figure 8 nanomaterials-15-01501-f008:**
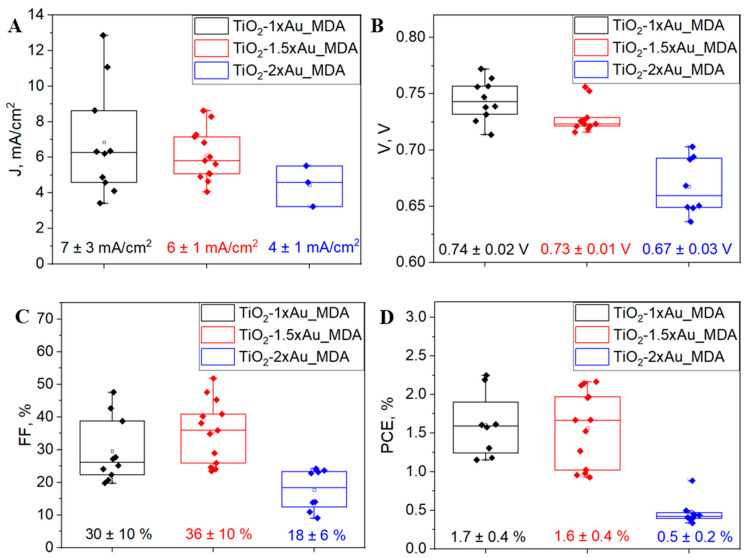
Photovoltaic performance parameters of PSCs incorporating TiO_2_-×Au_MDA layers with varying AuNP concentrations (1×, 1.5× and 2×). (**A**)—short circuit current density (Jsc), (**B**)—open circuit voltage (Voc), (**C**)—fill factor (FF), and (**D**)—power conversion efficiency (PCE) extracted from J-V measurements. The J-V curves are presented in Appendix A.

**Table 1 nanomaterials-15-01501-t001:** Photovoltaic performance of the PCE champion devices.

	Diameter of MDA, µm	J, mA/cm^2^ ± 0.1	V_OC_, V ± 0.01	PCE, % ± 0.3	FF, % ± 0.1
BL_A TiO_2_-Au_MDA	40	19.2	0.76	5.2	36.7
BL_F TiO_2_-Au_MDA	90	16.0	0.69	3.7	30.6
BL_P TiO_2_-Au_MDA	90	9.5	0.69	4.2	33.0

**Table 2 nanomaterials-15-01501-t002:** Photovoltaic performance of the PCE champion devices measured on day 1 and after 7 days of storage.

	Day of Measurement	J, mA/cm^2^ ± 0.1	V_OC_, V ± 0.01	FF, % ± 0.1	PCE, % ± 0.3
BL → TiO_2_-Au_MDA	1	18.6	0.73	25.6	3.5
	7	19.0	0.70	23.0	3.3
TiO_2_-Au_MDA → BL	1	10.0	0.70	20.0	1.0
	7	9.5	0.63	14.0	0.9

## Data Availability

The original contributions presented in this study are included in the article/Appendix A. Further inquiries can be directed to the corresponding author.

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
