# Peer review of "Systematic Study of Gold Nanoparticle Effects on the Performance and Stability of Perovskite Solar Cells"

_nanomaterials, 2025, doi:10.3390/nano15191501_

Round 1
Reviewer 1 Report
Comments and Suggestions for Authors
In this submission, the authors explore the integration of inkjet-printed TiO2-AuNP microdot arrays (MDA) into the electron transport layer to create a plasmonic interface for perovskite solar cells (PSCs). The manuscript presents a systematic study of the interface engineering strategy of AuNPs in PSCs, offering valuable insights for optimizing light capture and charge transport. While the overall framework is complete, there is potential for deeper analysis of certain results. Considering the high standards of the Nanomaterials, acceptance is recommended upon major revisions. To enhance the quality of their work, the authors are advised to address the following issues:
1. Figure 1 depicts a multiphase material, yet the distribution of different phases cannot be discerned from the morphology. It is suggested to include element distribution characterization.
2. Figure 2 appears to contradict Table 1. For instance, the FF of BL_F TiO2-Au_MDA is the highest in Figure 2 but the lowest in Table 1. Additionally, the author is requested to clarify the reasons for the varying VOC across different samples.
3. In Figure 2a, the red curve is partially obscured. Please readjust the vertical coordinate range to ensure all curves are clearly visible.
4. The performance degradation caused by high concentration (2×) is attributed to "aggregation". However, the TEM (Figure 6C) does not clearly demonstrate the aggregation phenomenon. It is recommended to incorporate quantitative data such as D50 and average particle size to better illustrate the aggregation phenomenon.
5. Determining the structural composition solely based on the XRD pattern is challenging. Please include the XRD standard card in Figure 7a for clearer identification.
6. Long-term stability is of significant importance for solar cells. It is advisable to supplement the manuscript with stability data.
7. Regarding AuNP concentration, "23 wt%" is mentioned in the abstract, while the main text refers to molar concentration (0.017M). It is suggested to uniformly use wt% throughout or provide the conversion method between the two.
Author Response
We would like to thank the reviewers for their constructive and very valuable comments. Let me address the issues raised by you one by one: Please note that the authors’ responses are in black, while the reviewers’ comments appear gray and italicized. Changes made to the text of the paper are highlighted in blue font. Reviewer #1: 1. Figure 1 depicts a multiphase material, yet the distribution of different phases cannot be discerned from the morphology. It is suggested that the element distribution be included. Thank you for this valuable suggestion. We agree that elemental mapping could provide complementary insights into the material distribution. However, due to the low concentration of Au within the scanned TiOâ‚‚ area, the error margin in EDS mapping is relatively high. Therefore, point EDS measurements were conducted instead of complete mapping. Following the reviewer's comment, we have added to Figure 1 the magnified micrographs using an additional backscattered electron (BSE) detector and included the EDS point analysis results as insets. The BSE micrographs, in particular, help visualize the Au distribution due to atomic number contrast, supporting the presence of gold within the microdot arrays. Additionally, the EDS mapping results for the AuNP have been included in the Supplementary Information (Figure S2). 2. Figure 2 appears to contradict Table 1. For instance, the FF of BL_F TiO2-Au_MDA is the highest in Figure 2 but the lowest in Table 1. Thank you for noticing this difference. Table 1 presents the photovoltaic parameters of the individual best-performing device for each architecture, selected to highlight the maximum achieved PCE values. However, devices demonstrating the highest PCE do not necessarily exhibit the highest value for each individual parameter, such as FF or Jsc.[ ] In contrast, Figure 2 shows the overall statistical distribution of all measured devices, reflecting the spread and average trends rather than isolated best performances. We have clarified this distinction in the revised manuscript to avoid potential confusion. 3. Additionally, the author is requested to clarify the reasons for the varying VOC across different samples. Our work exploits a standard spin-coating method for fabricating active layers in the devices. Namely, perovskites synthesized in solutions can have a number of defects, structural disturbances, and chemical heterogeneity, which can be recombination centers, and VOC is strongly affected by nonradiative recombination. Despite this, the loss of VOC caused by nonradiative recombination in the case of PSCs is slight compared to other solution-processed photovoltaic cells. In our experiments, the difference in VOC values within the same group of samples is comparable to the difference between groups of samples with various processing. We attribute this to the fact that inevitable errors in the preparation of samples in laboratory conditions and different BL treatments exert the main influence on the formation of defects. In particular, various defects can arise at the interfaces when spin-coating the perovskite and hole transport layers. Following the reviewer's note, we have clarified the reasons for the varying VOC across different samples in the “discussion” part of the manuscript. 4. In Figure 2a, the red curve is partially obscured. Please readjust the vertical coordinate range to ensure all curves are clearly visible. Thank you for your observation. Following the reviewer's comment, we have recalculated and corrected the vertical axis range in Figure 2a to ensure all the curves are visible. The revised Figure 2a now clearly presents all plotted spectra. 5. The performance degradation caused by high concentration (2×) is attributed to "aggregation". However, the TEM (Figure 6C) does not clearly demonstrate the aggregation phenomenon. It is recommended that quantitative data, such as D50 and average particle size, be incorporated to better illustrate the aggregation phenomenon. Thank you for your valuable comment. Figure 6C demonstrates that at the highest precursor concentration (2×), two distinct size populations of AuNPs are present. Alongside the evenly distributed smaller particles with an average size of (17±6) nm, larger particles with an average size of (51±5) nm appear. Indeed, the aggregation and size increase of nanoparticles with rising precursor concentration is a well-documented phenomenon. Specifically, the larger population results from accelerated growth kinetics at high precursor concentrations, followed by the coalescence of smaller particles into bigger aggregates. Additionally, very large nanostructures, approximately 250 nm in size, can also be observed within or surrounding the nanoparticle matrix. This latter observation further supports the occurrence of an agglomeration process. Following the reviewers' notes, we have added an extended discussion to clarify this point. Additionally, we have calculated D50 values based on Au NPS size distribution analysis. The D50 values are 35 nm for 1× concentration and 48 nm for 2× concentration. The notable increase in D50 for the 2× sample indicates a shift toward larger particle sizes, supporting our conclusion of increased aggregation or coalescence at higher concentrations. We added the D50 values to the manuscript and Figure 6D-F for clarity. 5. Determining the structural composition solely based on the XRD pattern is challenging. Please include the XRD standard card in Figure 7a for clearer identification. Thank you for your comment. We have added the XRD standard cards in Figure 7a and S7. Extended XRD spectra, including reference patterns for all relevant materials, have also been added to the Supplementary Information (Figure S7). 6. Long-term stability is of significant importance for solar cells. It is advisable to supplement the manuscript with stability data. Thank you for your valuable suggestion. We fully agree that long-term stability is critical for solar cell applications. While comprehensive extended stability tests are planned for future studies due to their time-consuming nature, we have included initial stability data in this manuscript. Specifically, to evaluate short-term stability, we conducted measurements after seven days of storage in a glove box, and the results have been added to the revised version (Table 2 and Table S2). 7. Regarding AuNP concentration, "23 wt%" is mentioned in the abstract, while the main text refers to molar concentration (0.017M). It is suggested to uniformly use wt% throughout or provide the conversion method between the two. Thank you for this valuable observation. For clarity, we have addressed your suggestion by adding the corresponding wt% of Au ions in the ink throughout the manuscript. It is important to note that these two types of concentration values describe different aspects of the process: The molar concentrations of 0.017 M (0.33 wt% Au ions in ink), 0.025 M (0.49 wt% Au ions in ink), and 0.034 M (0.65 wt% Au ions in ink) refer to the HAuClâ‚„ precursor content in the ink before printing. In contrast, the 23 wt%, 34.5 wt%, and 46 wt% values refer to the Au content relative to TiOâ‚‚ in the solid printed microdot arrays after annealing (i.e., solid-to-solid ratio in the final film). Both sets of values are important as they describe different preparation stages (ink formulation versus final solid composition) and are thus presented for completeness. We have clarified this distinction in the revised manuscript to avoid confusion.

Reviewer 2 Report
Comments and Suggestions for Authors
While the concept of incorporating inkjet-printed TiOâ‚‚-AuNP microdot arrays is interesting, the overall device efficiency remains too low to justify publication in its current form. The improvements reported are not sufficiently significant compared to the state-of-the-art, and concerns about reproducibility, interfacial defects, and scalability further undermine the practical value of this approach. Without demonstrating a clear and substantial efficiency gain, this work does not meet the standards required for acceptance.
1) Compared to the current state-of-the-art perovskite solar cells, the efficiency is too low to validate the overall concept.
2) There is no reference J-V data. The device without Au nanoparticles must be presented.
3) In the SEM iamges of Au nanoparticle, overall SEM image for TiO2 film should be presented.
Author Response
Response to the reviewer's comments
We would like to thank the reviewers for their constructive and very valuable comments. Let me address the issues raised by you one by one:
Please note that the authors' responses are in black, while the reviewers' comments appear gray and italicized. Changes made to the text of the paper are highlighted in blue font.
Reviewer #2:
1) Compared to the current state-of-the-art perovskite solar cells, the efficiency is too low to validate the overall concept.
The primary focus of this work is a systematic investigation of the effects of AuNP concentration, positioning, and substrate treatment on plasmonic enhancement and charge transport, rather than to maximize device performance. The fabrication conditions were intentionally kept constant without extensive optimization to isolate the studied effects. Future work will focus on integrating these findings into optimized PSC architectures to achieve higher efficiencies.
2) There is no reference J-V data. The device without Au nanoparticles must be presented.
Thank you for your comment. The J-V data for devices without Au nanoparticles (standard PSCs and PSCs with TiOâ‚‚ microdot arrays without gold) are now presented in the Supplementary information (Table S1, Figure S3-S7). We have also added an additional discussion on this point to the main text to direct readers to the reference data for comparison.
3) In the SEM images of the Au nanoparticle, the overall SEM image for the TiO2 film should be presented.
Thank you for your constructive suggestion. To address this, we have added optical microscopy images in the revised supplementary information (Figure S2), providing an overview of the printed microdot arrays on TiOâ‚‚ films with different treatments. We also added clarification to the text. These images complement the higher-magnification SEM and TEM images (Figures 2 and 6) by demonstrating the printed dots' overall distribution, uniformity, and morphology on the substrate and PCS samples.

Reviewer 3 Report
Comments and Suggestions for Authors
Title: Systematic study of gold nanoparticle effects on the performance and stability of perovskite solar cells
Journal: Nanomaterials
Manuscript number: nanomaterials-3734785
The manuscript presents findings from research on a plasmonic interface for perovskite solar cells (PSCs) through the integration of inkjet-printed TiO2-AuNP (gold nanoparticle) microdot arrays (MDA) into the electron transport layer. The authors systematically investigate how the conditioning of the TiO2 blocking layer (BL), the positioning of the AuNP layer, and the loading of nanoparticles collectively influence device performance. The topic addressed is both engaging and relevant to readers, and the subject matter is suitable for the scope of the journal, Nanomaterials.
However, several points require further clarification from the authors.
In details:
Figure 2a: The absorption spectra presented in Figure 2a are incorrect, as the absorption coefficient cannot be negative. It appears that the authors may not have calibrated the spectrophotometer properly before conducting the measurements. After correcting the absorption spectra, the authors should revise the corresponding text in the manuscript that discusses the spectral behavior (lines 162-191).
Figure 2, legend. There is a discrepancy between the figures and the description to them. For the TiO2 BL_P, the figure displays green curves, but the description states that these are blue curves.
Figure 7a. Authors should provide information regarding which peaks belong to which elements in the X-ray diffraction patterns. Without this information, the patterns lack interpretative value..
Conclusions. The authors do not offer any conclusions concerning the impact of nanoparticle sizes based on their results. The authors should correct this shortcoming.
Considering the above mentioned, I think the manuscript in its present form needs a major revision.
Author Response
Response to the reviewer's comments
We would like to thank the reviewers for their constructive and very valuable comments. Let me address the issues raised by you one by one:
Please note that the authors’ responses are in black, while the reviewers’ comments appear gray and italicized. Changes made to the text of the paper are highlighted in blue font.
Reviewer #3:
- Figure 2a: The absorption spectra presented in Figure 2a are incorrect, as the absorption coefficient cannot be negative. It appears that the authors may not have calibrated the spectrophotometer properly before conducting the measurements. After correcting the absorption spectra, the authors should revise the corresponding text in the manuscript that discusses the spectral behavior (lines 162-191).
Thank you for your observation. Following the reviewer's comment, we have recalculated and corrected the vertical axis range in Figure 2a to ensure all the curves are visible.
- Figure 2, legend. There is a discrepancy between the figures and the description of them. For the TiO2 BL_P, the Figure displays green curves, but the description states that these are blue curves.
Thank you for noticing this discrepancy. We have corrected the description to match the Figure, ensuring that the TiOâ‚‚ BL_P is consistently referred to with its appropriate color (green) throughout the manuscript.
- Figure 7a. Authors should provide information regarding which peaks belong to which elements in the X-ray diffraction patterns. Without this information, the patterns lack interpretative value.
Thank you for your comment. We added the XRD standard cards in Figure 7a and S7 to allow pick recognition. Extended XRD spectra, including reference patterns for all relevant materials, have also been added to the Supplementary Information (Figure S7).
- The authors do not offer any conclusions concerning the impact of nanoparticle sizes based on their results. The authors should correct this shortcoming.
Thank you for your comment. We added the conclusion about the impact of concentrations in the text and abstract.
Although higher AuNP concentrations improve dispersion stability, maintain MAPI crystallographic integrity, and yield more uniform nanoparticle size distributions, diode-like and photovoltaic measurements demonstrated that these advantages do not translate into better device performance. The 1x concentration provides optimal PSC efficiency by achieving the best balance between plasmonic enhancement and charge transport while avoiding the increased resistance, carrier scattering, and recombination losses observed at higher loadings.

Round 2
Reviewer 1 Report
Comments and Suggestions for Authors
This version can be accepted now.
Author Response
no comments
Reviewer 2 Report
Comments and Suggestions for Authors
Although the authors have sincerely responded to comments 2 and 3, the device efficiency (below 3%) still remains low compared to recent perovskite solar cells achieving almost 27%. Therefore, it is necessary not only to conduct a systematic investigation but also to improve performance aspects such as efficiency or stability in order to be published in a journal of Nanomaterials.
Author Response
comment:
- Although the authors have sincerely responded to comments 2 and 3, the device efficiency (below 3%) still remains low compared to recent perovskite solar cells, which have achieved almost 27%. Therefore, it is necessary not only to conduct a systematic investigation but also to improve performance aspects such as efficiency or stability in order to be published in a journal of Nanomaterials.
response:
We appreciate the reviewer’s feedback regarding device efficiency. We would also like to respectfully clarify that the primary focus of this study is not on maximizing absolute device efficiency, but rather on the fundamental investigation of plasmonic interface engineering using inkjet-printed TiOâ‚‚-AuNP microdot arrays (MDA). Our systematic approach provides insight into how AuNP positioning, concentration, and BL treatment influence light–matter interaction and charge transport, offering a foundation for future performance optimization. Further work will focus on translating these findings into high-efficiency PSCs.
Nevertheless, in response to the reviewer's comment, we have executed some preliminary optimizations of the devices and added Figure S8 to the Supplementary Information to demonstrate their performance. The devices, fabricated with a slightly optimized configuration, achieved power conversion efficiencies of up to ~8%. These results suggest that, upon further optimization of the fabrication conditions, the proposed plasmonic interface design is compatible with higher-performing device architectures.

Reviewer 3 Report
Comments and Suggestions for Authors
The authors have took into account all the reviewer's comments, and the revised manuscript can be accepted for publication.
Author Response
no comments